# Plastic sexual ornaments: Assessing temperature effects on color metrics in a color-changing reptile

**Braulio A. Assis** [1]*, **Benjamin J. M. Jarrett**[2], **Gabe Koscky**[3], **Tracy Langkilde**[1], **Julian D. Avery**[4]

**1** Department of Biology, Intercollege Graduate Degree Program in Ecology, The Pennsylvania State University, University Park, PA, United States of America, **2** Department of Entomology, Michigan State University, East Lansing, MI, United States of America, **3** Independent scholar, New York, NY, United States of America, **4** The Department of Ecosystem Science and Management, The Pennsylvania State University, University Park, PA, United States of America

\* bmd5458@psu.edu

**Data Availability Statement:** Data are available through Penn State's data repository, ScholarSphere, at https://doi.org/10.26207/01az-

## Abstract

Conspicuous coloration is an important subject in social communication and animal behavior, and it can provide valuable insight into the role of visual signals in social selection. However, animal coloration can be plastic and affected by abiotic factors such as temperature, making its quantification problematic. In such cases, careful consideration is required so that metric choices are consistent across environments and least sensitive to abiotic factors. A detailed assessment of plastic trait in response to environmental conditions could help identify more robust methods for quantifying color. Temperature affects sexual ornamentation of eastern fence lizards, *Sceloporus undulatus*, with ventral coloration shifting from green to blue hues as temperatures rise, making the calculation of saturation (color purity) difficult under conditions where temperatures vary. We aimed to characterize how abiotic factors influence phenotypic expression and to identify a metric for quantifying animal color that is either independent from temperature (ideally) or best conserves individual's ranks. We compared the rates of change in saturation across two temperature treatments using seven metrics: three that are based on fixed spectral ranges (with two of them designed by us specifically for this system) and three that track the expressed hue (with one of them designed by us to circumvent spurious results in unornamented individuals). We also applied a lizard visual sensitivity model to understand how temperature-induced color changes may be perceived by conspecifics. We show that the rate of change in saturation between two temperatures is inconsistent across individuals, increasing at a higher rate in individuals with higher baseline saturation at lower temperatures. In addition, the relative color rank of individuals in a population varies with the temperature standardized by the investigator, but more so for some metrics than others. While we were unable to completely eliminate the effect of temperature, current tools for quantifying color allowed us to use spectral data to estimate saturation in a variety of ways and to largely preserve saturation ranks of individuals across temperatures while avoiding erroneous color scores. We

0r85 R code is available at http://github.com/braulioassis/pavo.

**Funding:** Funding was provided in part by the National Science Foundation IOS-1456655 to T.L. The funders had no role in study design, data collection and analysis, decision to publish, or preparation of the manuscript.

**Competing interests:** The authors have declared that no competing interests exist.

describe our approaches and suggest best-practices for quantifying and interpreting color, particularly in cases where color changes in response to environmental factors.

## Introduction

Animal coloration is integral to visual communication channels and selective processes, and is an important subject of investigation in studies on social selection [1]. Coloration of conspicuous ornamental traits can predict the outcome of intrasexual competition and mate preference in many taxa [2–6], and for this reason is expected to be under strong selection. Notably, animal coloration can be plastic, as rapid changes in hue may occur in response to environmental conditions such as temperature, hydration, and background coloration, particularly among ectotherms [7–13]. Such dynamic changes make these color traits an important target of investigation on their roles in social interactions and signaling potential [14], but this same complexity brings challenges to researchers: to carefully consider optimal color quantification methods that account for plasticity, and to accurately estimate signal strength when color states are variable. One alternative is to standardize the abiotic conditions that are most representative of natural settings in which color traits may influence animal interactions. However, this is not always possible (e.g., when measuring traits under field conditions) or ecologically realistic (e.g., when thermoregulation influences fitness via multiple routes). Another option would be to develop a metric that is completely independent from abiotic factors, or at least conserves ranks of color of individuals across that environmental gradient. The latter would be sufficient to answer several important biological questions related to other qualitative fitness correlates such as mate choice [5,15] and social hierarchy [16,17] and may thus be a valid effort.

Unlike pigment-based coloration, structural colors typically arise from the reflection of light on nanostructures located inside iridophores [18–20] or on matrices of keratin [13] and chitin [21]. The conformation of such structures may respond to changes in temperature and osmolarity [22], altering the wavelengths reflected and consequently the hue perceived by the receiver. Due to these characteristics, color traits that are structural in nature may demand more attention when determining methods and environments for their quantification. To fully capture the color properties of such complex traits, researchers often employ spectrophotometry, as it can generate raw spectral data that is not biased towards the human visual system [23]. However, this approach alone does not inform us about how a given signal is perceived by potential receivers. Models that account for visual sensitivity of signal receivers are being more commonly employed [24,25] but are not the norm. A detailed assessment of how different color metrics and visual models behave across environmental treatments could provide guidelines of best practices for working with organisms that exhibit color plasticity and how these changes may be perceived by target receivers.

Eastern fence lizards, *Sceloporus undulatus*, display conspicuous color patches on their ventral gular and abdominal regions. These are present in sexually mature males and seem to be relevant in visual displays during social interactions [26]. During encounters with rivals and potential mates, male fence lizards often perform push-up displays, elevating their bodies and revealing their ventral coloration [27,28]. For this reason, it is likely that ventral color functions as a badge of quality [29] that may influence the outcome of these competitive encounters leading to individual differences in fitness. Among females, this trait can be expressed to varying degrees, albeit less conspicuously than in males, or be entirely absent [30]. Moreover, variation in color appears to be fitness-relevant for this species: males with more saturated badges

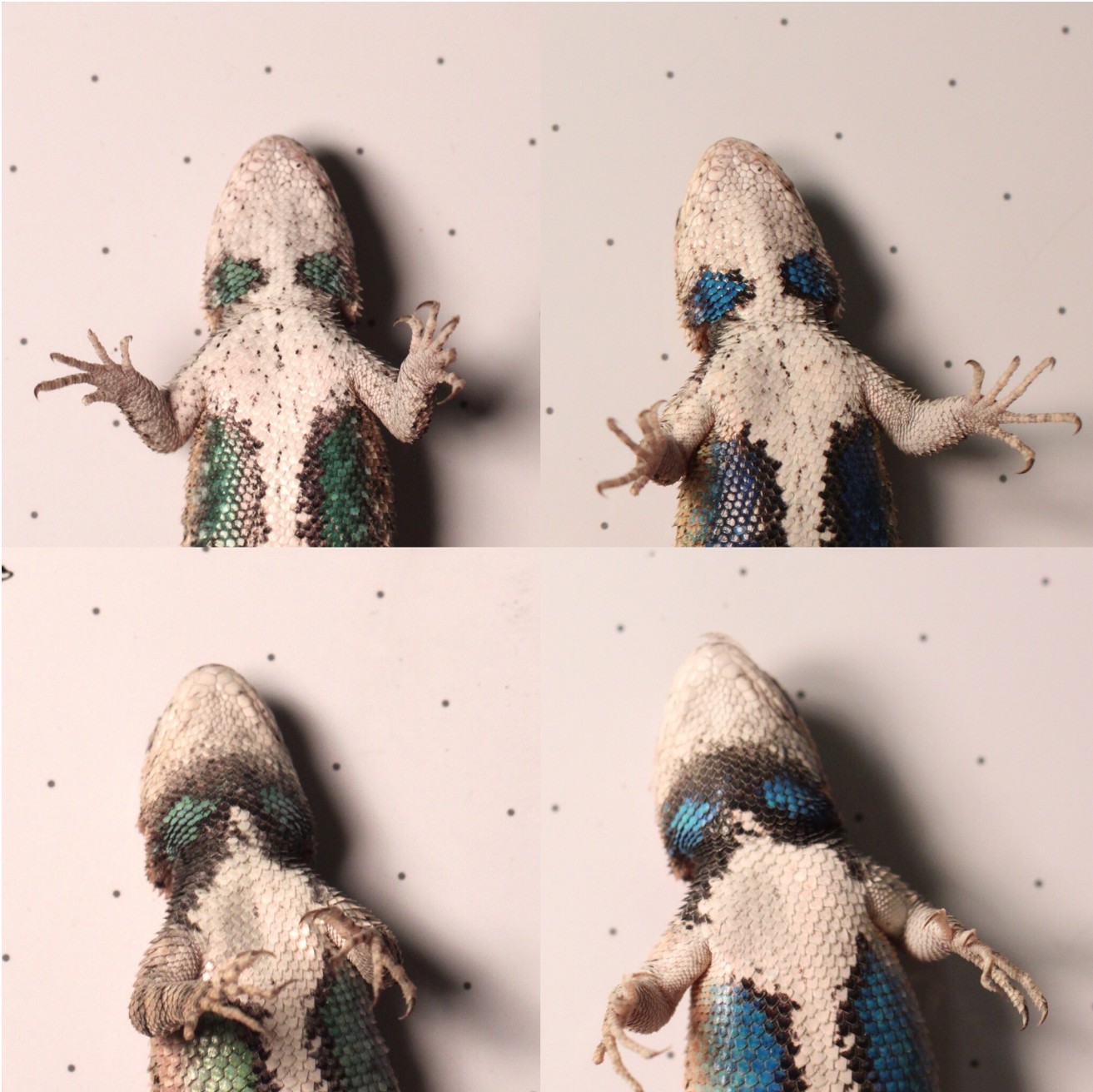

**Fig 1. Two male lizards (top and bottom) at ~23˚C (left panels) and ~33˚C (right panels).**

are more likely to be larger-bodied [31], and males with larger badges are preferred by females [32]. On the other hand, females exhibiting color are less preferred by males [30], but achieve faster running speeds and have offspring that evade predatory attacks more often [33].

The structural ventral coloration in *S. undulatus* displays pronounced plasticity in hue, shifting from medium wavelengths (green) to shorter wavelengths (blue) with increasing temperatures [34] (Fig 1). Because of this shift, it is unlikely that hue alone signals an individual's inherent quality. Hue may, however, signal an individual's current thermal state, which has

fitness consequences in competitive interactions [35,36]. Saturation, on the other hand, is referred to in color science as the general appearance of a given color and specifies the size of the difference from the most similar achromatic color (i.e. gray) [37]. Here we define saturation as a measure of the relative purity of a color signal, typically calculated as the ratio of light reflectance of a specific hue to light reflectance of the full spectral range. Therefore a highly saturated color patch reflects light in a constrained range of the spectrum, with fewer wavelengths outside of that range contributing to that signal. Highly saturated colors tend to indicate purer and more vivid signals, and many studies in animals have shown a link between saturation and diet quality, resource availability, or immunocompetence [38–41]. However, because saturation is often calculated as relative reflectance in a specific spectral range, this metric can be problematic if the predominant spectral range (hue) is variable across individuals or responds to abiotic factors.

To further explore color quantification methods in organisms exhibiting environment-dependent color plasticity, we compared seven metrics of saturation under two temperature treatments relevant to the ecology of *S. undulatus*. Three of these metrics are derived over fixed spectral ranges, with two of them on customized spectral ranges that better suit *S. undulatus* coloration. Three others are flexible and track the expressed hue [36,42–45]. Finally, to understand how this signal (and its plasticity) may be perceived by conspecifics, we employed a visual model based on the spectral sensitivities of a lizard species, *Crotaphytus dickersonae* [46]. We explain these metrics in more detail below (see *Color metrics*). Our objectives were to 1) determine whether badge hue and saturation vary at consistent rates across individuals between different temperature environments, 2) determine which measure of saturation best preserves an individual's color rank across temperatures, and 3) provide direction for how to better measure and consider the role of coloration in animal behavior studies when the focal species exhibits phenotypic color plasticity.

## Methods

### Study organism

Lizards were raised in the lab from eggs (15 females and 17 males) that were obtained from gravid females collected at field sites in Tennessee (Land Between the Lakes National Recreation Area, and Edgar Evins State Park) and Arkansas (Mississippi River State Park, and private lands in Lee County). Juveniles were housed in groups of no more than five non-siblings in plastic containers (45 x 30 x 25cm) provided with a 15 x 15cm piece of opaque corrugated plastic to be used as shelter, and a lamp with a 60W incandescent bulb suspended 25cm above one end of the container and turned on daily from 08:00 to 16:00 to allow thermoregulation. Animals were fed *Acheta domesticus* crickets three days per week, one day of which was supplemented with Reptivite reptile vitamins (Zoo Med Laboratories Inc., San Luis Obispo, CA, USA). A small dish containing water was available to animals at all times. The room was maintained at a temperature of 23°C and a 12:12 light cycle from overhead lights.

### Color quantification

Color measurements were taken at a mean age of 338.1 ± 2.7 days, which is sufficient for *S. undulatus* to reach reproductive maturity [47]. Before measurement, individuals were acclimated in an incubator (Quincy Lab, Chicago, IL, USA) until the targeted internal body temperature for each treatment was reached (22.9 ± 0.24°C for the *cold* treatment, and 32.9 ± 1.4°C for the *warm* treatment). Body temperatures were assessed by inserting a Fluke Bead Probe connected to a Thermocouple thermometer (Fluke Corporation, Everett, WA, USA) into an individual's cloaca. To quantify color, we used an Ocean Optics Jaz UV/VIS

spectrometer with a pulsed xenon light source to measure reflectance. Spectra were taken perpendicular to the subject and calculated relative to a diffuse white standard (Ocean Optics WS-1) using SpectraSuite. We measured reflectance of the colored portion of the lizard's left throat badge, with an integration time of 40μs and a trigger period of 10μs. Each individual's badge was measured three times, by removing and replacing the probe each time within the colored region. For each of the three measurements, SpectraSuite took 5 scans and averaged them to produce one spectrum. We then used the R package *pavo* [48] version 2.1.0 to interpolate each spectrum to 1 nm intervals and used the *procspec* function to smooth the spectra with a span of 2/3 [49]. The directionality of temperature shifts influences the speed in which hues change [50], and for this reason all individuals were measured in the cold treatment first. In order to investigate the full range of color variability in this system, we also included in our sample females bearing no ornamentation, in which case the background coloration in the same region of the throat was measured instead.

## Color metrics

We used the '*summary.rspec*' function of the *pavo* R package [48] with custom modifications to the code to extract a total of six saturation metrics (Table 1). Our modified code is available at https://github.com/braulioassis/pavo/. The metrics represented fixed and flexible spectrum ranges, and comprised metrics already established in the literature and others designed by us specifically for this system. They were: *S1B* (fixed on the "blue" portion of the spectrum); *S1Sc* (fixed on the "turquoise" range of the spectrum, better aligned with the spectral reflectance of *S. undulatus* badges); *S1ScFull* (encompassing the full range of hue fluctuation from blue to green); *S3* (flexible and bracketing maximum reflectance); *S8* (flexible and contrasting maximal and minimal reflectance); and *S3Sc* (same as *S3*, but modified to not allow the center of the range to be greater than 600 nm). This last metric was of particular interest to us, since our sample contained females that were weakly ornamented. These females exhibited peaks of reflectance at up to 700 nm, beyond the scope of the ornament, generating a saturation value that was derived from the individual's background coloration rather than ornamentation

**Table 1. Summary of saturation metrics.** Metrics in bold are not generated by *pavo* automatically and required changes in the code (available at http://github.com/braulioassis/pavo).

|  | Metric | Description | Formula |
|---|---|---|---|
| Fixed | *S1B* | "Blue" range of the spectrum, 400 to 510 nm | $\dfrac{\int_{400}^{510} R(\lambda)d\lambda}{\int_{300}^{700} R(\lambda)d\lambda}$ |
|  | **S1Sc** | "Turquoise" range of the spectrum, 450 to 550 nm | $\dfrac{\int_{450}^{550} R(\lambda)d\lambda}{\int_{300}^{700} R(\lambda)d\lambda}$ |
|  | **S1ScFull** | Full range from "blue" to "green", 400 to 600 nm | $\dfrac{\int_{400}^{600} R(\lambda)d\lambda}{\int_{300}^{700} R(\lambda)d\lambda}$ |
| Flexible | *S3* | On the range of peak reflectance ± 50 nm | $\dfrac{\int_{\lambda_{max}-50}^{\lambda_{max}+50} R(\lambda)d\lambda}{\int_{300}^{700} R(\lambda)d\lambda}$ |
|  | **S3Sc** | Same as S3, but not centered beyond 600 nm | $\dfrac{\int_{\lambda_{max}-50}^{\lambda_{max}+50} R(\lambda)d\lambda}{\int_{300}^{700} R(\lambda)d\lambda}$, <br> $if\ \lambda_{max}>600\rightarrow set\ \lambda_{max}=600$ |
|  | *S8* | Difference between maximal and minimal reflectance | $\dfrac{\max(R)-\min(R)}{\int_{300}^{700} R(\lambda)d\lambda}$ |

All metrics are calculated in relation to total reflectance from 300 to 700 nm. λ: wavelength; R: percent reflectance at a given λ; $\lambda_{max}$: λ of maximal R. The sum of the reflectances over a range $[\lambda_1, \lambda_2]$ is the equivalent of the area under the curve for that range, which can be expressed as the integral $\int_{\lambda_1}^{\lambda_2} R(\lambda)d\lambda$.

(Fig 2). By constraining the center of the target range to be no more than 600 nm, we attempted to capture the most relevant information tracking the blue to green fluctuations in ornamented individuals while ensuring a low saturation values for weakly ornamented females, this way preventing false ornamentation scores derived from background coloration. All metrics were calculated in relation to total brightness, defined as the total spectral reflectance from 300 to 700 nm. In addition, we analyzed the rate of change in the ornament's hue for males and females by measuring the wavelength of maximal reflectance (*H1*) at each temperature treatment. Lastly, to obtain a complete picture of color dynamics of this species, we also analyzed changes in mean brightness (mean reflectance at all wavelengths of the spectral range) across the two temperature environments. This way, we present results encompassing three important characteristics of color: hue, saturation, and brightness [45].

For each individual in a treatment, the seven saturation metrics were derived from the same spectra, eliminating error that could arise from unique readings for each of the metrics. It is worth mentioning that *pavo* generates several other metrics [45] not evaluated by us in this study. We chose metrics that were robust to spectral noise and relevant to the hues of our study organism, but other metrics may be better suited to other systems. Additionally, fence lizard badges do not exhibit iridescence, and for this reason we do not discuss issues in color quantification that arise with this phenomenon, such as angle of measurement. Useful insight on the subject is available elsewhere [51].

## Visual model

The spectral sensitivity model was built using the function *sensmodel* in the R package *pavo* [48]. Cone sensitivities have not yet been determined for *S. undulatus*, so we applied visual parameters established for the closest related iguanid, *Crotaphytus dickersonae* (Crotaphytidae): 359, 459, 481, 558 [46]. Cone type ratios were assumed to be even. The sensitivity model was applied to the spectral data using the function *vismodel* and projected in a tetrahedral space using the function *colspace*. Saturation was determined irrespective of the expressed hue by the length of the vector *r*, corrected for the maximum length of *r* for that hue projected in a non-spherical volume, *r.achieved* [24].

## Statistical analyses

All analyses were done in R 3.5.0 [52] using the *lme4* package [53]. All three measures of an individual's badge were included in each analysis with the random term of ID accounting for the lack on independence. To test whether the change in hue from green to blue with increasing temperatures is consistent across sex, we fit a mixed model (model 1: *metric ~ temperature * sex + (temperature | ID) + $\varepsilon$*, where $\varepsilon$ is the error term) where each individual had a unique intercept (wavelength of maximal reflectance, or *H1*, at the mean temperature) and a unique slope (the rate in which *H1* decreases with temperature). Sex was included as an interaction with temperature (which was centered on the mean). We then fit the same model but without the interaction between sex and temperature (model 2: *metric ~ temperature + sex + (temperature | ID) + $\varepsilon$*) and compared the two using a Chi-square test to determine which model best explained the data. If this interaction best explained the data, we split the data by sex and analyzed them separately. We included both a random intercept and random slope for each individual in both models as the primary reason was to assess sex differences in their reaction norms, and this model best reflected the plotted data.

For the models where each sex was analyzed separately, we constructed first a simpler model than model 2 by constraining individuals to only have a random intercept (model 3: *metric ~ temperature + (1 | ID) + $\varepsilon$*). Model 4 (*metric ~ temperature + (temperature | ID) + $\varepsilon$*)

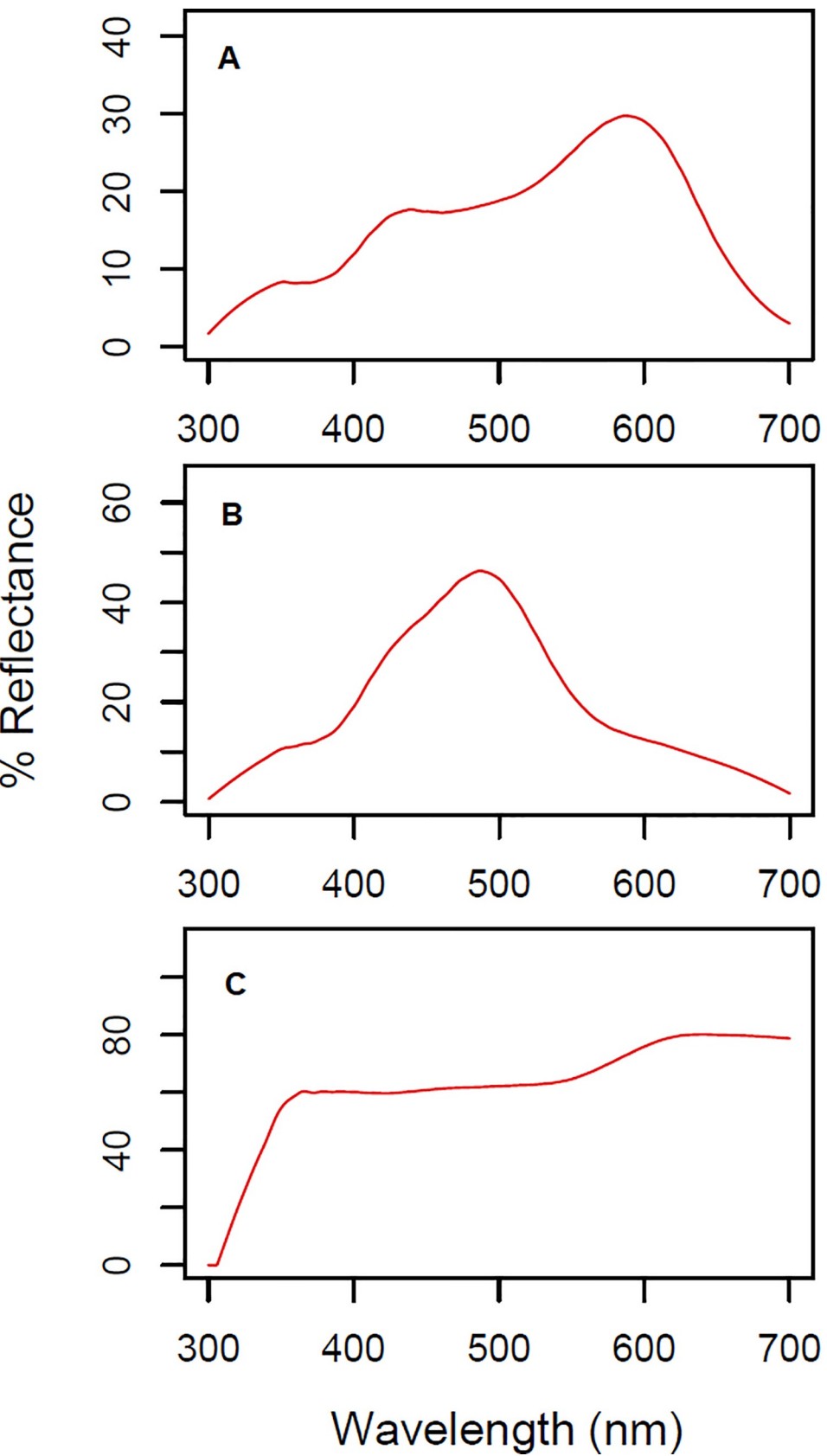

**Fig 2. Spectrum profiles (% reflectance) of throat badges representative of our sample.** Hue (*H1*) is equal to the wavelength of maximal reflectance. Saturation is calculated as reflectance at a defined range in relation to reflectance at the "full" range (established as 300–700 nm in our study). A. male individual at ~23°C; B. same individual as in 'A', but at ~33°C; C. weakly ornamented female, with greatest reflection at wavelengths corresponding to background coloration.

allowed both a random intercept and random slope for each individual, which is analogous to model 2, but was only performed on each sex in turn. Comparing model 3 and model 4 allowed us to determine whether individuals change hue at the same rate across temperatures. If model 4 best explained the data, then individuals do vary in their reaction norm, in which case we would fit a fifth model (model 5: *metric ~ temperature + (temperature || ID) + ε*). Model 4 allowed the intercept and slope of each individual to covary. Model 5 constrains the correlation between the slope and intercept. If model 4 best explains the data, this would mean the slope and intercept are significantly correlated. If model 5 best explains the data, there is no correlation between the slope and the intercept of an individual. A positive slope-intercept correlation means that individuals that exhibit, in this case, a high *H1* baseline level, also increase *H1* at a greater rate across the temperature environment. Next, to analyze the reaction norms of color saturation with respect to temperature, we replicated the same analysis for each of seven saturation metrics: *S1B, S3, S8, S1Sc, S1ScFull, S3Sc,* and the visual sensitivity model. This allowed us to test whether saturation increases consistently across individuals in a similar way. We ascertained the 95% confidence intervals around the correlation using the *confint* function in *lme4* which would allow qualitative comparison between the seven saturation metrics. Saturation was $\log_e$-transformed prior to analysis and temperature was included as a fixed term and centered on the mean so the intercept is at the mean temperature. In addition, we performed a Spearman rank order correlation test as a second method to explore which of the six metrics best preserves the individuals' relative saturation ranks across cold and warm temperatures.

We assessed the repeatability for all seven saturation metrics (measured in triplicate) using the R package *rptR* [54]. We tested the repeatability for a metric within each treatment based on a normal distribution with 1000 parametric bootstraps.

### Ethics statement

The research presented here adhered to Guidelines for the Use of Animals in Research, the legal requirements of the U.S.A. and the Institutional Guidelines of The Pennsylvania State University and was approved by IACUC. Animal collection was authorized by the respective states' permits.

### Results

All saturation calculations derived from spectral data were highly repeatable (for *cold* treatment: all r > 0.808, p < 0.001; for *warm* treatment, all r > 0.782, p < 0.001). Receptor excitation for the four cone types estimated by the visual sensitivity model are presented in Table 2. A two-dimensional projection of the tetrahedral color space illustrates how cone excitation differed between the two temperature environments for all individuals (Fig 3).

There were significant sex differences (all P < 0.001; see Table 3) in the reaction norms of all saturation indices but not for *H1* (hue, or wavelength of maximal reflectance, $\chi^2_1 = 0.86$, P = 0.35) or *B2* (mean brightness, $\chi^2_1 = 2.41$, P = 0.12). We therefore split the female and male data and analyzed them separately for all indices except *H1* and *B2* which were analyzed with males and females grouped.

**Table 2. Mean (± standard deviation) receptor excitation for the four cone types for males and females across the two temperature treatments.**

| Sex | Treatment | U | S | M | L |
|---|---|---|---|---|---|
| F | Cold | 0.195 ± 0.012 | 0.253 ± 0.006 | 0.259 ± 0.005 | 0.292 ± 0.011 |
| F | Warm | 0.164 ± 0.013 | 0.268 ± 0.009 | 0.284 ± 0.008 | 0.284 ± 0.016 |
| M | Cold | 0.118 ± 0.027 | 0.266 ± 0.009 | 0.284 ± 0.011 | 0.331 ± 0.025 |
| M | Warm | 0.061 ± 0.043 | 0.334 ± 0.032 | 0.353 ± 0.029 | 0.252 ± 0.030 |

F: females, n = 15; M: males, n = 17; Cold: 22.9 ± 0.24˚C; Warm: 32.9 ± 1.4˚C; U: 359 nm; S: 459 nm; M: 481 nm, L: 558 nm.

The temperature treatment was effective in altering the hue of lizards' badges (*cold* treatment: 600 ± 35 nm; warm treatment: 502 ± 21 nm; paired t-test: t = 34.6, df = 95, P < 0.001). All females in the *cold* treatment exhibited peak reflectance at wavelengths greater than 600 nm, indicating that background hues were predominant in individuals without pronounced ornamentation. At warmer temperatures, however, badges showed increased reflectance at shorter wavelengths in relation to background hues, and all peaks occurred below 600 nm. The reaction norms for *H1* were best explained by model 4: individuals have unique slopes ($\chi^2_2$ = 79.90, P < 0.001) but there is no correlation between the intercept and the slope of each individual ($\chi^2_1$ = 0.05, P = 0.83, corr = -0.04, 95% confidence intervals [-0.40, 0.36]). This means that the hue at one temperature does not predict the hue at the other.

We observed different patterns for all other metrics of saturation (Table 3, Fig 4). As for *H1*, all metrics showed the same pattern of individuals having unique slopes (model 4 best explained the data over model 3, all P < 0.001). As all individuals had unique intercepts and slopes, we looked for significant correlations between intercepts and slopes. With random slopes, observing a phenotype at one temperature might not provide information on the phenotype in a different environment. If the slope and intercept are correlated, a phenotype in one environment may convey some information that would allow a prediction of that same phenotype in another environment. By comparing model 4 and model 5, we found significant correlations between slope and intercept for males and females in all metrics (Table 3) except females when looking at *S1ScFull* ($\chi^2_1$ = 2.57, P = 0.11, corr = 0.43 [-0.09, 0.83]). The significant correlations varied from -0.82 [-1.00, -0.22] (*S3* in females) to 1.00 [0.82, 1.00] (*S8* in males). A positive correlation means that individuals that have a higher intercept have a greater

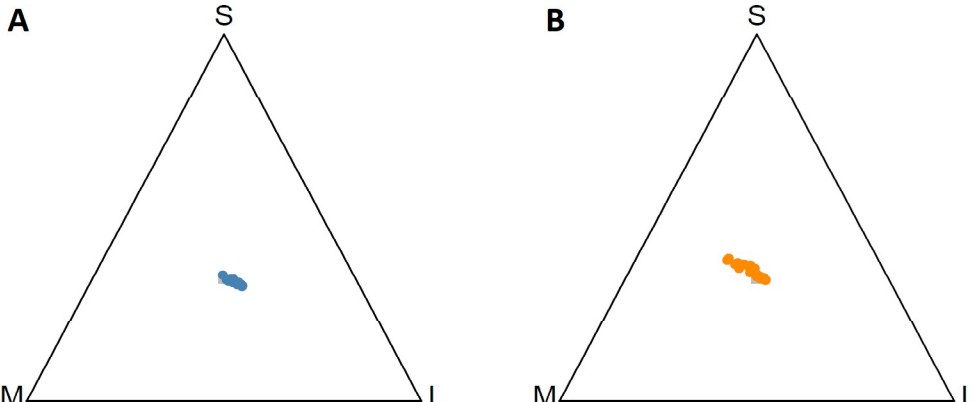

**Fig 3. Tetrahedral color spaces observed from above the UV vertex indicating cone excitation for all individuals across two environments.** A: *cold* treatment, 22.9 ± 0.24˚C; B: *warm* treatment, 32.9 ± 1.4˚C; S: small cone type (peak sensitivity at 459 nm); M: medium cone type (peak sensitivity at 481 nm); L: long cone type (peak sensitivity at 558 nm).

**Table 3. Results from all statistical tests for seven saturation metrics (Table 1).** If model 1 and model 2 are significantly different, it indicates that the sexes have different reactions norms. If model 3 and model 4 are significantly different, it means that individuals differ in their slopes. If model 4 and 5 are significantly different it means that the correlation between an individual's slope and intercept is also significant. The slope-intercept correlation is the estimated correlation between an individual's intercept and the slope of their reaction norm, with 95% confidence intervals.

| Metric | Sex | Comparison of model 1 and model 2 | Comparison of model 3 and model 4 | Comparison of model 4 and model 5 | Slope-intercept correlation | Spearman rank correlation |
|---|---|---|---|---|---|---|
| S1B | M | | $\chi^2_2 = 71.42$, P < 0.001 | $\chi^2_1 = 16.19$, P < 0.001 | 0.85 [0.60, 0.98] | S = 514, P = 0.36, ρ = 0.244 |
| | F | $\chi^2_1 = 20.48$, P < 0.001 | $\chi^2_2 = 71.10$, P < 0.001 | $\chi^2_1 = 8.53$, P = 0.003 | 0.69 [0.18, 0.91] | S = 468, P = 0.56, ρ = 0.164 |
| S1Sc | M | | $\chi^2_2 = 53.00$, P < 0.001 | $\chi^2_1 = 9.55$, P = 0.002 | 0.73 [0.33, 0.95] | S = 420, P = 0.14, ρ = 0.38 |
| | F | $\chi^2_1 = 15.04$, P < 0.001 | $\chi^2_2 = 101.03$, P < 0.001 | $\chi^2_1 = 14.40$, P < 0.001 | 0.82 [0.47, 0.96] | S = 616, P = 0.72, ρ = -0.1 |
| S1ScFull | M | | $\chi^2_2 = 33.97$, P < 0.001 | $\chi^2_1 = 10.80$, P = 0.001 | 0.79 [0.39, 1.00] | S = 306, P = 0.03, ρ = 0.55 |
| | F | $\chi^2_1 = 10.93$, P < 0.001 | $\chi^2_2 = 57.56$, P < 0.001 | $\chi^2_1 = 5.77$, P = 0.016 | 0.61 [0.11, 0.89] | S = 366, P = 0.21, ρ = 0.35 |
| S3 | M | | $\chi^2_2 = 44.52$, P < 0.001 | $\chi^2_1 = 23.82$, P < 0.001 | 0.99 [0.85, 1.00] | S = 168, P < 0.01, ρ = 0.75 |
| | F | $\chi^2_1 = 16.36$, P < 0.001 | $\chi^2_2 = 17.20$, P < 0.001 | $\chi^2_1 = 6.95$, P = 0.008 | -0.82 [-1.00, -0.22] | S = 404, P = 0.31, ρ = 0.28 |
| S3Sc | M | | $\chi^2_2 = 44.58$, P < 0.001 | $\chi^2_1 = 23.79$, P < 0.001 | 0.99 [0.85, 1.00] | S = 168, P < 0.01, ρ = 0.75 |
| | F | $\chi^2_1 = 30.72$, P < 0.001 | $\chi^2_2 = 29.47$, P < 0.001 | $\chi^2_1 = 2.57$, P = 0.11 | 0.43 [-0.09, 0.83] | S = 260, P = 0.04, ρ = 0.54 |
| S8 | M | | $\chi^2_2 = 39.04$, P < 0.001 | $\chi^2_1 = 21.27$, P < 0.001 | 1.00 [0.82, 1.00] | S = 186, P < 0.02, ρ = 0.73 |
| | F | $\chi^2_1 = 22.99$, P < 0.001 | $\chi^2_2 = 58.61$, P < 0.001 | $\chi^2_1 = 4.14$, P = 0.04 | 0.53 [0.01, 0.86] | S = 566, P = 0.97, ρ = -0.01 |
| Visual model | M | | $\chi^2_2 = 29.78$, P < 0.001 | $\chi^2_1 = 9.26$, P = 0.002 | 0.75 [0.35, 0.99] | S = 208, P = 0.04, ρ = 0.69 |
| | F | $\chi^2_1 = 11.45$, P < 0.001 | $\chi^2_2 = 36.14$, P < 0.001 | $\chi^2_1 = 1.17$, P = 0.28 | 0.30 [-0.31, 0.76] | S = 218, P = 0.02, ρ = 0.61 |

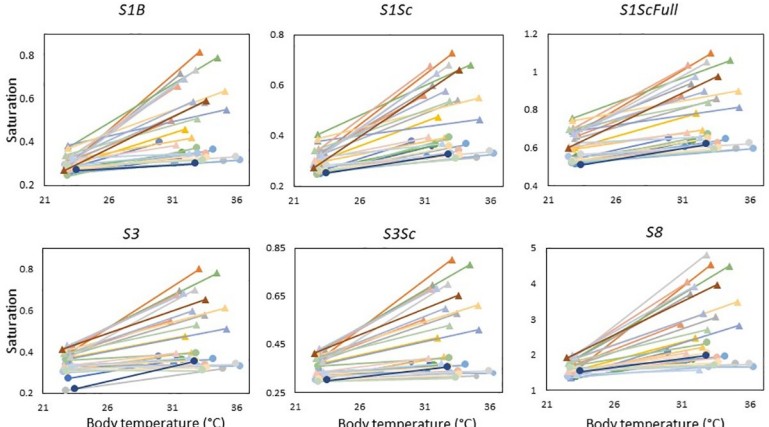

**Fig 4. Reaction norms for saturation from six spectral ranges measured at two temperature treatments.** Male lizards are represented as triangles, and females as circles. Metrics are summarized in Table 1. All metrics were calculated in relation to total reflectance over the 300–700 nm spectral range.

slope, and so that metric increases more rapidly across temperature. A negative correlation (only observed for *S3* in females) means than individuals with larger intercepts have shallower slopes, and so that metric changes less across the temperature.

The rank order correlation (Fig 5) adds a different perspective in that it does not take into account the shift in phenotype and asks only whether an individual has a greater saturation value at both temperatures. The rank order correlation was significant for males when looking at *S3* (S = 168, P = 0.001, rho = 0.75) and *S8* (S = 186, P = 0.002, rho = 0.73), and for both males (S = 168, P = 0.001, rho = 0.75) and females (S = 260, P = 0.04, rho = 0.54) when looking at *S3Sc*.

Finally, the analysis of mean brightness (*B2*) showed that even though individuals have different slopes ($\chi^2_2$ = 40.29, P < 0.001), there is no evidence for a correlation between slope and intercept ($\chi^2_1$ = 0.19, P = 0.66; corr = 0.09 [-0.31, 0.50]. This indicates that, even though the badges of *S. undulatus* change in brightness at different rates between individuals, the rate of change and its directionality are not easily predicted by basal levels of brightness, making it a less comparable metric.

## Discussion

In order to equate plastic phenotypic traits to individual fitness it is important to first understand how abiotic factors influence phenotypic expression. We show that sexually-selected badges in eastern fence lizards (*Sceloporus undulatus*) are affected by increasing temperatures, by not only shifting hue from green to blue [34,50], but also by increasing the saturation of the color signal. More importantly, the magnitude of the effect of temperature on saturation is not consistent across sexes or individuals. For all metrics tested (with the exception of *S3Sc* and the visual model, for females), saturation increased at significantly higher rates in males and in

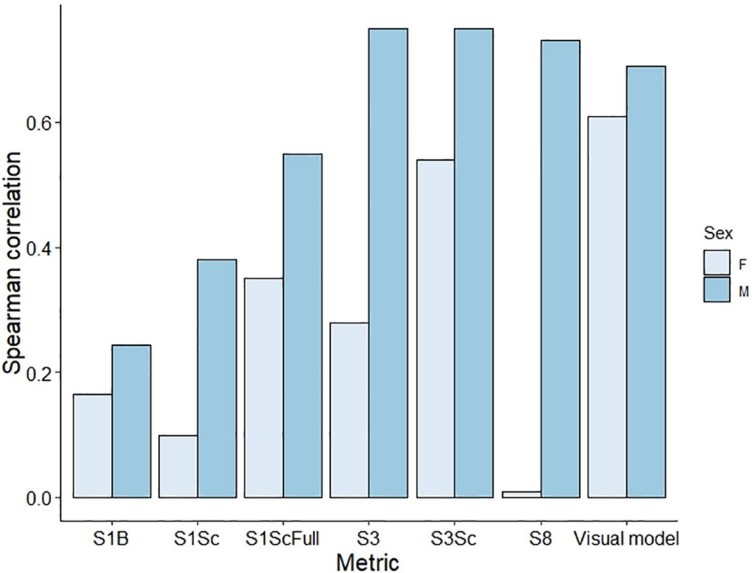

**Fig 5. Spearman correlation scores of males and females between two temperature treatments and calculated using seven saturation metrics.** The first three metrics are calculated from fixed spectral ranges, whereas the following three use spectral ranges that track the expressed hue. *Visual model* corresponds to saturation values irrespective of hue and based on visual sensitivity parameters established for *Crotaphytus dickersonae*. For ornamented males, scores are identical between *S3Sc* (a custom metric) and the standard *S3* metric generated by *pavo* [48]. When females are included, some of which are weakly ornamented, *S3Sc* provides the highest correlation across the two temperatures.

individuals with higher baseline color saturation when cold. These results highlight how even accounting for abiotic factors statistically may not completely remove the noise caused by color plasticity, but some metrics may preserve relative ranks from individuals more consistently than others. Since spectrophotometry allows researchers to derive a variety of metrics from single measurements [45,48] and thus does not require separate handlings of samples and animals for new measurements, we suggest that researchers are attentive to these alternatives. Even if color traits do not exhibit plasticity, it may be worthwhile to verify whether hue distributions fall neatly within the fixed spectral brackets established, and if not, to consider metrics based on fluctuating spectral brackets instead. Finally, we attempted to objectively quantify color as well as its absence in unornamented individuals using a single, objective metric that does not rely on human visual judgment. Below, we further discuss our metric assessment and considerations in the *S. undulatus* system.

For the badges of *S. undulatus*, we show that the relative strength of a color signal across individuals depends upon the standardized temperature at which they were measured. This issue could be potentially minimized by selecting a standard measuring temperature that would be most representative of natural conditions for the studied population, or of a time of day in which social interactions, and consequently signaling displays, may be predominant. For example, relative color ranks at cooler temperatures as those seen in periods of low lizard activity may not be informative of their significance in competitive outcomes, and therefore should be disfavored. Secondly, we showed that the degree of plasticity itself varies across individuals within and between sexes and is correlated with baseline levels of saturation. In this case, temperature becomes a less reliable predictor of color saturation across a heterogenous sample and accounting for it statistically may not be sufficient in providing an accurate picture. In our study, a sampling regime across a more comprehensive temperature gradient would have allowed us to capture the thermal sensitivity of this trait in much more detail–and obtaining reaction norms for various degrees of ornamentation across the two sexes could have been the ideal way to calculate an optimal correction for temperature. Because such reaction norms may be logistically challenging to determine for this and other systems, however, a metric that best preserves individuals' relative ranks may be sufficient in reducing statistical noise while requiring significantly less effort. Finally, our metric comparison highlights how weakly ornamented individuals can generate spurious measures of saturation when using flexible metrics, as background hues may be scored instead. In our case, this issue was significantly diminished by preventing the focal spectral range from moving beyond the hue range of the ornament.

Relative color ranks of individuals were the least consistent across temperature treatments when saturation was based on fixed spectral ranges. Among these, the most direct approach of quantifying relative reflectance of "blue" wavelengths (*S1B*, [45]) showed the lowest correlation across temperatures for males. Designing fixed-range metrics tailored to our study system, however, improved correlations of saturation across the two temperatures. After evaluating both the distribution of expressed hues in our sample and its range of plasticity, we chose to calculate saturation in spectral ranges different from those predetermined and that better suited this trait. *S1ScFull* (400–600 nm) outperformed *S1Sc* (450–550 nm) and *S1B* (400–510 nm) across temperature treatments for both males and females in this regard. We employed *S1ScFull* in an attempt to cover the full range of hue fluctuation in *S. undulatus*, but this strategy introduces challenges of its own: by calculating saturation over such a broad spectral range (400–600 nm, or 50% of the designated full range of 300–700 nm), individuals exhibiting high reflectance at a wide range of wavelengths (and therefore a more diluted, less pure color closer to white) would have an overestimated saturation compared to individuals reflecting narrowly defined wavelengths in either the blue or green ranges. Consequently, we are faced with the

challenge of establishing a target range narrow enough to accurately represent color purity, but wide enough to encompass the species' range of plasticity.

Calculating saturation over ranges which track fluctuations in hue preserved the relative color ranks of individuals significantly better across treatments, with the bracketing of peak reflectance at ± 50 nm (*S3*, as in [44]) performing marginally better in males than when contrasting maximum and minimum reflectance (*S8*, as in [36, 43,55, 56]). The former approach may also be more biologically relevant because cone pigment spectral sensitivities in animal visual systems are approximately Gaussian and have a half width of roughly 100 nm [57]. In females, however, correlations between treatments were low, particularly for *S8*. Since we were interested in capturing the full range of phenotypic variation in this species, we aimed for a metric that would accurately differentiate between ornamented and unornamented individuals, rather than providing false scores based on background color saturation. In the absence of ornamentation, background body coloration reflects longer wavelengths (> 600 nm) that can generate high saturation scores not pertaining to the ornament itself (since it is absent). By modifying the *S3* metric and preventing the spectral range from being centered beyond 600 nm, we designed a new metric (*S3Sc*) that had a correlation of saturation across temperatures identical to those from the *S3* for males, but that also had the highest correlation for females. Our results suggest that for study systems in which the color traits exhibit hue plasticity and are present in all individuals, the standard *S3* metric generated by *pavo* may be the most robust. However, if the absence of color is also of interest, modifications may be needed so that background coloration is not confounded with the focal ornament.

It is important to note, however, that we are not certain whether a uniform metric that accurately characterizes color richness as well as its absence is an achievable goal in a system so complex and dynamic as this. An alternative to be considered would be to manually score unornamented individuals with "zeroes". An individual may be objectively determined as unornamented if its spectral curve exhibits peak reflectance outside the range of the trait found in ornamented individuals. Rather than calculating saturation at the expected range for an absent trait, assigning such cases as zeroes should be a simpler, and arguably, equivalent solution. Here we chose to evaluate and present a method that is in fact informed directly by the spectral curves and is better encased in the population's data range and distribution. Therefore, of all the options considered by us, *S3Sc* (flexible range, but only within the scope of the trait) appears to be the best approach.

Quantifying color using raw spectral reflectance may provide us with information that is not biased towards any particular species' visual sensitivity, and therefore give us a more accurate insight about resource allocation (for example, pigments or structures [58–60]) into color development irrespective of how that signal is perceived by other organisms [61]. Still, in order to understand the role of color signals in social interactions and competition, it may be more informative to assess color through the lens of the signal receiver [62]. For this purpose, visual sensitivity models have been frequently employed in an effort to filter out reflectance signatures that are irrelevant to the sensory system of the target. Unfortunately, determining peak sensitivities for each cone type as well as cone type ratios for all systems is a challenging task, and no sensitivity parameters have been established for phrynosomatid lizards so far. Still, we applied a sensitivity model based on a crotaphytid lizard [46] assuming a well-conserved visual system across iguanians [63,64]. Like other saturation metrics that track the expressed hue rather than focus on a set spectral range, this approach allows us to estimate saturation (the corrected vector length from the origin) irrespective of hue changes (vector angle changes that represent hues) with temperature variation. However, similar patterns were observed with this approach, in which saturation exhibited a positive correlation between slope and intercept across temperature treatments. Spearman rank correlations for saturation scores across

treatments were not as high as *S3* or *S3Sc* for males but were the highest for females. It appears that accounting for lizard visual sensitivity increased the consistency of color ranks for weakly-saturated individuals. However, since unornamented females were included in the analysis, some of these color scores correspond not to the trait itself, but to background coloration in the trait's absence.

Considerations on the plastic nature of *S. undulatus* badges can be made not only from a quantitative perspective, but also on the ecological role of this signal in a social context. The saturation plasticity detected by us contributes to the hypothesis that badges in *S. undulatus* function as a signal of immediate competitive ability affected by body temperature [34]. In nature, individuals at warmer temperatures should exhibit more saturated, bluer hues and this would consequently advertise their greater physical performance associated with warmer body temperatures [35,36]. Future studies on the role of variation in color and temperature on social interactions could provide important insight into the social role of this trait.

Rapid color changes are a known phenomenon across many ectotherms, and even though the dramatic changes seen in *S. undulatus* may be rare, the points discussed here may be relevant to other systems. Pacific tree frogs (*Hyla regilla*) exhibit complex patterns of color change that involve the interaction of multiple environmental factors such as temperature and background coloration [65]. Standardizing multiple abiotic conditions can be difficult, but careful consideration of color metric alternatives may prove helpful. Some species of fish also display fast color changes [66] or alternate color states that correlate with dominance status, such as in the cichlid *Astatotilapia burtoni* [67]. Males of this species can be non-territorial and cryptically colored or be territorial and exhibit alternate morphotypes of blue or yellow coloration. Quantifying color uniformly in such a system may be as challenging as in *S. undulatus*, but here we have discussed some options that could be considered.

Visual signals can be dynamic and complex. Field ecologists normally face logistical challenges that affect the level of control achievable when sampling in natural settings, such as seasonality, time of day, and the measurement of populations across latitudinal clines [68]. Under such circumstances, the potential for error and noise in data collection can be increased if the trait of interest is plastic. In such cases it is advantageous to establish a metric that measures this signal and is robust to abiotic variation. We demonstrate that quantification of sexual color saturation in *S. undulatus* is strongly influenced by temperature and illustrate how assessing color in organisms that exhibit this phenomenon requires careful consideration. Importantly, the effect of temperature on saturation is not consistent across all individuals, but rather it increases in magnitude with increasing baseline saturation levels. Among our candidate metrics, the one that best characterized this plastic coloration trait was saturation bracketing maximum reflectance via full-spectrum spectrometry. However, if the absence of the targeted trait is also of interest, our modifications to that metric should be useful to avoid confounding dominant background coloration with ornaments. Employing visual sensitivity models are important improvements, but unfortunately many systems still do not have their visual parameters determined. We have shown that observing color dynamics of a study organism can improve its accuracy, for example by defining custom spectral ranges that better suit a given system. Our modifications to the code of *pavo* [48] illustrate such examples and could perhaps be implemented in future versions of the package (e.g. user-defined spectral ranges for calculation of saturation). Overall, these observations illustrate how color assessment should be made in a careful manner that accounts for the targeted trait's characteristics, and identifying appropriate methods for doing such will be critical if we are to advance our knowledge on the role visual signals in animal behavior and fitness.

## Acknowledgments

We would like to thank G. McCormick and C. Tylan for assistance in field work, the staff at Edgar Evins State Park, Standing Stone State Park, and Land Between the Lakes National Recreation Area in Tennessee, and at Mississippi River State Park in Arkansas. We are most grateful for the Lansdale family in allowing us to collect individuals in their property. We thank E. Mainou and two anonymous reviewers for critical assistance and comments on the manuscript. We would also like to express our greatest appreciation for H. I. Engler and her wholehearted commitment to animal care and husbandry. The authors declare no conflict of interest.

## Author Contributions

**Conceptualization:** Braulio A. Assis, Benjamin J. M. Jarrett, Julian D. Avery.

**Data curation:** Braulio A. Assis, Benjamin J. M. Jarrett, Gabe Koscky.

**Formal analysis:** Benjamin J. M. Jarrett, Gabe Koscky.

**Funding acquisition:** Tracy Langkilde.

**Investigation:** Braulio A. Assis, Julian D. Avery.

**Methodology:** Braulio A. Assis, Gabe Koscky, Julian D. Avery.

**Project administration:** Braulio A. Assis, Julian D. Avery.

**Resources:** Tracy Langkilde.

**Software:** Braulio A. Assis, Gabe Koscky.

**Supervision:** Braulio A. Assis, Julian D. Avery.

**Writing – original draft:** Braulio A. Assis, Benjamin J. M. Jarrett, Tracy Langkilde, Julian D. Avery.

**Writing – review & editing:** Braulio A. Assis, Benjamin J. M. Jarrett, Gabe Koscky, Tracy Langkilde, Julian D. Avery.

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
