## [Decision Letter · Decision Letter 0]

25 Nov 2019

PONE-D-19-26164

Plastic sexual ornaments: metric choice for quantifying coloration in color-changing animals

PLOS ONE

Dear Mr Assis,

Thank you for submitting your manuscript to PLOS ONE. Both reviewers liked to work, and I agree that it is potentially very but there are concerns which at present mean that it does not fully meet PLOS ONE’s publication criteria as it currently stands. Therefore, we invite you to submit a revised version of the manuscript that addresses each the points raised  by both referees. Please send a covering letter explaining your responses to each point.

ould appreciate receiving your revised manuscript by Jan 09 2020 11:59PM. To enhance the reproducibility of your results, we recommend that if applicable you deposit your laboratory protocols in protocols.io, where a protocol can be assigned its own identifier (DOI) such that it can be cited independently in the future. For instructions see: http://journals.plos.org/plosone/s/submission-guidelines#loc-laboratory-protocols

We look forward to receiving your revised manuscript.

Kind regards,

Daniel Osorio

Academic Editor

PLOS ONE

Journal Requirements:

Reviewers' comments:

Reviewer's Responses to Questions

**Comments to the Author**

1. Is the manuscript technically sound, and do the data support the conclusions?

Reviewer #1: Yes

Reviewer #2: Partly

2. Has the statistical analysis been performed appropriately and rigorously? 

Reviewer #1: Yes

Reviewer #2: No

3. Have the authors made all data underlying the findings in their manuscript fully available?

Reviewer #1: No

Reviewer #2: Yes

4. Is the manuscript presented in an intelligible fashion and written in standard English?

Reviewer #1: Yes

Reviewer #2: Yes

5. Review Comments to the Author

Reviewer #1: In this study, the authors try out different colourimetric indices to quantify hue and saturation of a temperature dependent signal in the Eastern fence lizard. Since different studies measure colour at different temperatures (especially on the field where this parameter cannot always be controlled), the values of these indices might not be comparable across studies. Here, the authors try to find out which index varies the least with temperature, or at least, which one does not change the individuals ranks. This manuscript is well written and a valuable contribution. I think it will be worth publishing once some details are made clearer/fixed, and once it is better explained how this study answers the question raised in the introduction.

# Major comments

- I think the work done here is a valuable enterprise but I'm a bit confused since at the end of the manuscript, I'm not entirely convinced you have manage to solve the problem mentioned in the introduction. Indeed, none of the metrics you proposed deals with the issue of different studies measuring at different temperature. Plus, there is no proof the work you did here can be transposed to other plastic ornaments. Maybe this can be solved by framing the article a bit less as a methodological article and insist more on the biological results:

* in the introduction, explain that even though a metric independent from temperature would be ideal, a metric that conserves ranks is sufficient to answer many biological questions (provide a few examples of questions where it would be the case)

* insist more on the (in my opinion) main result on this study: saturation (measured with appropriate metrics) ranks are well conserved across temperatures in Eastern fend lizards (and possibly use this result as your title instead of a methodological question that is not answered with certainty here)

- I don't really understand how limiting the range to 300-600 nm (S3Sc) can help not capturing the effect of the background colouration. From what I understand looking at the Figure 2 (c panel), you will always measure R[550-650]/B2 since ""H1"" (not really H1 actually since this metric was designed to be used on bell-shaped spectra) will always be 600nm. This doesn't seem to provide any useful information about the saturation. This needs more thought but I'm even convinced it is possible to design a metric that works (and makes sense) for both ornamented and unornamented individuals... This needs more discussion in this introduction and the discussion itself.

# Line by line comments:

- L44-45: see Shawkey et al. (2011) 10.1016/j.zool.2010.11.001 for an additional example of the effect of hydration on colour, in a bird this time

- L54: add a ref and a sentence to explain what 'badge' means in the context of selection selection

- L71-72: refs? Here are some but they mainly concern iridescent colours in other taxa. It would be better if you could provide some studies on non-iridescent colours in lizards:

1. Fitzstephens DM, Getty T. Colour, fat and social status in male damselflies, Calopteryx maculata. Animal Behaviour. 1 Dec 2000;60(6):851‑5.

2. Meadows MG, Roudybush TE, McGraw KJ. Dietary protein level affects iridescent coloration in Anna’s hummingbirds, Calypte anna. Journal of Experimental Biology. 15 Aug 2012;215(16):2742‑50.

- L86: is there a specific reason you didn't test other saturation metrics from Montgomerie (2006) such as S2 or S4? If so, it would be useful to explain these reasons.

- L120: please add the exact version number you used for pavo. This will help with reproducibility in the future in case some functions change. Also please cite the article (10.1111/2041-210X.13174) instead of the package (or cite both if you prefer).

- L121: this part is a bit confused. The interpolation done by pavo (performed during the file import step by `getspec()` or class conversion step by `as.rspec()`) is a linear interpolation. It is often recommended to smooth the spectra afterwards, especially for metrics sensitive to noise such as H1, but this is done by the `procspec()` function. Looking at Figure 2, the spectra don't seem smoothed. If you actually did use smoothing (which is absolutely crucial here in my opinion), please precise the span value used (as recommended in White et al. 2015 10.1016/j.anbehav.2015.05.007) and please update your plots in Figure 2 (it makes much more sense to show the data used to compute the indices instead of the raw data).

- L130: the code has not been posted on GitHub yet, which means it is not available for review. If I get this manuscript for a second round of review, I would really like to see the code.

- L133-137: please provide the formula or a figure that illustrates each one of these indices. Currently, it would be hard to understand for someone who hasn't read Montgomerie (2006)

- L152: did you measure repeatability for each one of these indices? Especially for the "new" ones (S3Sc in particular which is the most unusual one). It's also not clear what happened to the 3 measurements you did on each individual. Did you average them for the rest of your analysis?

- L169-174: I usually find it more legible/hepful to include the formula of your model rather than a description text. Could you please do both if you'd like to keep the text?

- L221: your spectral data is most likely interpolated and pruned at every nm (it is for sure if you use pavo), so it doesn't make much sense to include decimals in your average and sd.

- L239: S3ScFull is not defined in table 1. Is this a typo?

- L242-243: it would be helpful to more clearly say that the only negative correlation is S3 in females (you say L240 it is negative but it's not clear it's the only one).

- L328: I'm not a big fan of the term 'lower vertebrates' which convey the illusion of a directional evolution (the term 'invertebrates' itself is borderline since it's not a monophyletetic group but it's more common and less damaging).

# Minor general comments:

- add a picture of the investigated badge somewhere. Maybe it's your Fig. 1? This figure is missing from the pdf I got

- I would like to see more discussion about how this could (or not) be applied to other organisms displaying temperature dependent colouration or plastic ornaments in general

Reviewer #2: The authors of "Plastic sexual ornaments: metric choice for quantifying coloration in color-changing animals" (PONE-D-19-26164) present the results of a study designed to uncover i) how color changes as a consequence of temperature in eastern fence lizards and ii) how the metrics chosen to summarize relevant attributes of the ornamental colors can interact with the temperatures at which color measurements are taken to influence the conclusions drawn from such measures. This is an interesting study and the authors draw several important conclusions: First, the study further highlights the importance of standardizing the measurement temperature for species known to exhibit thermally-dependent color change. Second, the authors highlight the finding that even standardizing a given temperature can have important implications regarding color-based phenotypic comparisons (e.g. individual A may be more colorful than individual B at low and intermediate temps, but less colorful than individual B at high temps).

My thoughts about the manuscript fall into two categories. First, though I think that the conclusions mentioned earlier are valuable and agree that these points should be made, I am not convinced that the experimental design adequately captures the relevant variation in color as a consequence of thermal sensitivity needed to put this variability in the appropriate context. Specifically, the authors discuss the thermal responsiveness of color in the context of reaction norms, yet measure and compare the color of their lizards at only two temperatures (though there are more than two temperatures represented collectively, each lizard has only two color values at each of two temperatures). Simply put, this sampling regime does not provide adequate insight into the thermal responses of the lizards. A more comprehensive sampling regime across temperatures might enable the collection and analysis of the data using well-developed techniques for analyzing and comparing reaction norms. Summary metrics of the reaction norms of each individual might therefore be the more appropriate way to deal with the temperature-sensitivity of color rather than looking for specific, and somewhat contrived, metrics of color that reproduce the ‘rank’ of individuals across temperatures and colors. This is the angle I excitedly anticipated based on the title and early parts of the paper, and I think such an approach could still be quite valuable with the right dataset.

Second, given that these color badges are frequently studied in the context of social signaling, the fact that these analyses focused exclusively on the spectral characteristics of the colors measured without any account of the spectral sensitivities of the relevant receivers was quite surprising (and an oversight, in my opinion). There is a deep and growing literature on the visual capabilities of many animals, and interpreting signals within the appropriate, receiver-dependent context provides important context and insight into the selective forces generating and maintaining diversity in signaling traits.

6. PLOS authors have the option to publish the peer review history of their article (what does this mean?). If published, this will include your full peer review and any attached files.

Reviewer #1: No

Reviewer #2: No

---

## [Author Response · Author response to Decision Letter 0]

26 Feb 2020

February 25, 2020

Dr. Daniel Osorio

Editor

PLOS One

Dear Dr. Osorio,

Thank you for the insightful reviews of our manuscript “Plastic sexual ornaments: metric choice for quantifying coloration in color-changing animals” [PONE-D-19-26164]. We appreciate the time and effort you and the reviewers put into assessing this manuscript. We have made the suggested changes to the manuscript, including implementing a visual sensitivity model in our data, and believe these have greatly improved its quality. We hope that you now find this acceptable for publication. Our responses to the reviewer’s comments are given below.

Sincerely,

Braulio Assis

Reviewer: 1 

Major comments: - I think the work done here is a valuable enterprise but I'm a bit confused since at the end of the manuscript, I'm not entirely convinced you have manage to solve the problem mentioned in the introduction. Indeed, none of the metrics you proposed deals with the issue of different studies measuring at different temperature. Plus, there is no proof the work you did here can be transposed to other plastic ornaments. Maybe this can be solved by framing the article a bit less as a methodological article and insist more on the biological results:

* in the introduction, explain that even though a metric independent from temperature would be ideal, a metric that conserves ranks is sufficient to answer many biological questions (provide a few examples of questions where it would be the case)

* insist more on the (in my opinion) main result on this study: saturation (measured with appropriate metrics) ranks are well conserved across temperatures in Eastern fend lizards (and possibly use this result as your title instead of a methodological question that is not answered with certainty here)

 Response: we appreciate these suggestions and have reframed parts of the Abstract, Introduction and Discussion to better focus on the goal of conserving ranks in saturation across temperatures and its applicability to important biological questions in future studies. 

- I don't really understand how limiting the range to 300-600 nm (S3Sc) can help not capturing the effect of the background colouration. From what I understand looking at the Figure 2 (c panel), you will always measure R[550-650]/B2 since ""H1"" (not really H1 actually since this metric was designed to be used on bell-shaped spectra) will always be 600nm. This doesn't seem to provide any useful information about the saturation. This needs more thought but I'm even convinced it is possible to design a metric that works (and makes sense) for both ornamented and unornamented individuals... This needs more discussion in this introduction and the discussion itself.

 Response: these concerns are valid and have been subject to serious considerations by us. Ideally, we would have liked to design a metric that captured color purity across the full gradient of variability within a population, which normally includes unornamented individuals. One simple solution to this problem would have been to assign scores of zero to individuals with peak reflectances at wavelengths outside the scope of the color trait (i.e. > 600). However, we decided against this option mainly in an idealistic effort to derive a metric that, is truly informed by the data, provides a meaningful and accurate value, and stays true to the definition of saturation itself: relative reflectance at a targeted spectral range. Preventing the focal range from fluctuating above 600nm was our best solution to this problem – the focal range remains within the limits of the color trait, is able to track green and blue hues expressed by ornamented individuals, and in the case of unornamented lizards still is appropriately scored as “low reflectance” in relation to high reflectances at λ > 600. Although we agree that caution should be exercised with these interpretations, we do believe that, given the circumstances, this may be the best option available (besides perhaps assigning artificial scores of zero). We argue that the information provided is actually useful – after all, the metric still represents relative reflectance at the targeted range for the trait, but without the risk of wrongfully targeting background coloration. In light of your concerns we have more carefully discussed these caveats in the updated manuscript.

- L44-45: see Shawkey et al. (2011) 10.1016/j.zool.2010.11.001 for an additional example of the effect of hydration on colour, in a bird this time

 Response: we have included this reference in the revised manuscript.

- L54: add a ref and a sentence to explain what 'badge' means in the context of selection

 Response: we removed the term “badge” from that line and only use it again later in the manuscript after introducing and referencing the concept (in line 97).

- L71-72: refs? Here are some but they mainly concern iridescent colours in other taxa. It would be better if you could provide some studies on non-iridescent colours in lizards:

1. Fitzstephens DM, Getty T. Colour, fat and social status in male damselflies, Calopteryx maculata. Animal Behaviour. 1 Dec 2000;60(6):851‑5.

2. Meadows MG, Roudybush TE, McGraw KJ. Dietary protein level affects iridescent coloration in Anna’s hummingbirds, Calypte anna. Journal of Experimental Biology. 15 Aug 2012;215(16):2742‑50.

Response: several references have now been included for this statement, most of which are on non-iridescent colors in lizards.

- L86: is there a specific reason you didn't test other saturation metrics from Montgomerie (2006) such as S2 or S4? If so, it would be useful to explain these reasons.

 Response: We chose those metrics that were least sensitive to noisy spectra, in part because we did not average the three scans into one measurement per individual. The text has been updated to reflect this reasoning, and we clarify to the reader that other non-tested metrics from Montgomerie (2006) are available in pavo and may be considered for other systems.

- L120: please add the exact version number you used for pavo. This will help with reproducibility in the future in case some functions change. Also please cite the article (10.1111/2041-210X.13174) instead of the package (or cite both if you prefer).

 Response: the package version and the correct citation for the Methods Ecol. Evol. article are now specified.

- L121: this part is a bit confused. The interpolation done by pavo (performed during the file import step by `getspec()` or class conversion step by `as.rspec()`) is a linear interpolation. It is often recommended to smooth the spectra afterwards, especially for metrics sensitive to noise such as H1, but this is done by the `procspec()` function. Looking at Figure 2, the spectra don't seem smoothed. If you actually did use smoothing (which is absolutely crucial here in my opinion), please precise the span value used (as recommended in White et al. 2015 10.1016/j.anbehav.2015.05.007) and please update your plots in Figure 2 (it makes much more sense to show the data used to compute the indices instead of the raw data).

 Response: Spectra were interpolated to 1nm intervals during file import and then smoothed when we called the procspec() routine. We used “smooth” with a span of 2/3. The manuscript text now reflects these choices.

- L130: the code has not been posted on GitHub yet, which means it is not available for review. If I get this manuscript for a second round of review, I would really like to see the code.

 Response: The forked version is now available on GitHub. You can find the full patch here: https://github.com/braulioassis/pavo/commit/cc964a79bc2683bd336e75d1198b570959b70a35

- L133-137: please provide the formula or a figure that illustrates each one of these indices. Currently, it would be hard to understand for someone who hasn't read Montgomerie (2006)

 Response: we have added a mathematical description for all metrics to Table 1.

- L152: did you measure repeatability for each one of these indices? Especially for the "new" ones (S3Sc in particular which is the most unusual one). It's also not clear what happened to the 3 measurements you did on each individual. Did you average them for the rest of your analysis?

 Response: we now report repeatability results for all metrics. All were satisfactorily repeatable, including S3Sc. We clarify that all of the three measurements on each individual were included in all analyses, with the individual added as a random effect. 

- L169-174: I usually find it more legible/hepful to include the formula of your model rather than a description text. Could you please do both if you'd like to keep the text?

 Response: we have added formulas to the text description of all models.

- L221: your spectral data is most likely interpolated and pruned at every nm (it is for sure if you use pavo), so it doesn't make much sense to include decimals in your average and sd.

 Response: averages and SD’s are now expressed as integers.

- L239: S3ScFull is not defined in table 1. Is this a typo?

 Response: that was indeed a typo and is now corrected.

- L242-243: it would be helpful to more clearly say that the only negative correlation is S3 in females (you say L240 it is negative but it's not clear it's the only one).

 Response: we have now clarified that this was the only negative correlation observed.

- L328: I'm not a big fan of the term 'lower vertebrates' which convey the illusion of a directional evolution (the term 'invertebrates' itself is borderline since it's not a monophyletetic group but it's more common and less damaging).

 Response: we removed the term “lower vertebrates” from the manuscript.

# Minor general comments:

- add a picture of the investigated badge somewhere. Maybe it's your Fig. 1? This figure is missing from the pdf I got

 Response: We apologize that the images were not available as they help to interpret our findings. Photos of lizards showing different colors had been included, but some error in the submission process may have occurred. We will be certain that they are visible in the revised manuscript. 

- I would like to see more discussion about how this could (or not) be applied to other organisms displaying temperature dependent colouration or plastic ornaments in general

 Response: These are great suggestions, and we now discuss parallels that can be made between our system and other groups such as fishes and amphibians that also exhibit color change. 

Reviewer #2: The authors of "Plastic sexual ornaments: metric choice for quantifying coloration in color-changing animals" (PONE-D-19-26164) present the results of a study designed to uncover i) how color changes as a consequence of temperature in eastern fence lizards and ii) how the metrics chosen to summarize relevant attributes of the ornamental colors can interact with the temperatures at which color measurements are taken to influence the conclusions drawn from such measures. This is an interesting study and the authors draw several important conclusions: First, the study further highlights the importance of standardizing the measurement temperature for species known to exhibit thermally-dependent color change. Second, the authors highlight the finding that even standardizing a given temperature can have important implications regarding color-based phenotypic comparisons (e.g. individual A may be more colorful than individual B at low and intermediate temps, but less colorful than individual B at high temps).

My thoughts about the manuscript fall into two categories. First, though I think that the conclusions mentioned earlier are valuable and agree that these points should be made, I am not convinced that the experimental design adequately captures the relevant variation in color as a consequence of thermal sensitivity needed to put this variability in the appropriate context. Specifically, the authors discuss the thermal responsiveness of color in the context of reaction norms, yet measure and compare the color of their lizards at only two temperatures (though there are more than two temperatures represented collectively, each lizard has only two color values at each of two temperatures). Simply put, this sampling regime does not provide adequate insight into the thermal responses of the lizards. A more comprehensive sampling regime across temperatures might enable the collection and analysis of the data using well-developed techniques for analyzing and comparing reaction norms. Summary metrics of the reaction norms of each individual might therefore be the more appropriate way to deal with the temperature-sensitivity of color rather than looking for specific, and somewhat contrived, metrics of color that reproduce the ‘rank’ of individuals across temperatures and colors. This is the angle I excitedly anticipated based on the title and early parts of the paper, and I think such an approach could still be quite valuable with the right dataset.

Response: Agreed. In hindsight, it would have been useful to employ a comprehensive temperature regime that could more accurately inform us of color change dynamics in this species. By attempting to demonstrate the drastic changes in color and sampling in more extreme temperature treatments, we failed to consider that data from intermediate points would be useful in determining a way of accounting for its own effect (if anything, at least for this species). While a greater range of temperatures would be ideal, we believe that we make a valuable contribution by bringing important points to discussion for color research and providing creative ideas for circumventing these issues. We feel that discussing our results in the context of reaction norms is appropriate and now emphasize that we analyzed these data in a reaction-norm framework. This was in combination with the analysis of rank order correlations to provide a two-pronged analytic approach to these data.

Second, given that these color badges are frequently studied in the context of social signaling, the fact that these analyses focused exclusively on the spectral characteristics of the colors measured without any account of the spectral sensitivities of the relevant receivers was quite surprising (and an oversight, in my opinion). There is a deep and growing literature on the visual capabilities of many animals, and interpreting signals within the appropriate, receiver-dependent context provides important context and insight into the selective forces generating and maintaining diversity in signaling traits.

 Response: Even though the signaling potential of the trait was not our initial focus for this project, your comment convinced us that applying a visual sensitivity model would provide important insight on how color plasticity could be perceived by conspecifics. Moreover, we agree that projecting color data in a color space – and consequently extracting the length of the vector from the origin irrespective of its direction in hue – is a very useful way of quantifying color purity independently of the expressed hue that was initially overlooked by us. Although visual sensitivity parameters have not yet been determined for this family of lizards, we still decided to follow this suggestion and presented a new analysis using data from such a model using parameters from a close relative. We thank you for this valuable suggestion which we believe has greatly improved the manuscript.

---

## [Editor Report · Decision Letter 1]

11 Mar 2020

PONE-D-19-26164R1

Plastic sexual ornaments: assessing temperature effects on color metrics in a color-changing reptile

PLOS ONE

Dear Mr Assis,

Thank you for submitting your manuscript to PLOS ONE. After careful consideration, we feel that it has merit but does not fully meet PLOS ONE’s publication criteria as it currently stands. Therefore, we invite you to submit a revised version of the manuscript that addresses the points raised during the review process.

We would appreciate receiving your revised manuscript by Apr 25 2020 11:59PM. To enhance the reproducibility of your results, we recommend that if applicable you deposit your laboratory protocols in protocols.io, where a protocol can be assigned its own identifier (DOI) such that it can be cited independently in the future. For instructions see: http://journals.plos.org/plosone/s/submission-guidelines#loc-laboratory-protocols

We look forward to receiving your revised manuscript.

Kind regards,

Daniel Osorio

Academic Editor

PLOS ONE

Additional Editor Comments (if provided):

This ms. makes a useful contribution to the literature on colour signalling and colour measurement, it is thoughtful and will be appreciated by many in the field. I judge also that the main conclusions are sound, and am happy to recommend publication in PLoS One. I do however have a number of comments which you might consider.

1. I am not aware of any other example of an animal showing such dramatic colour changes with temperature at least in a what is clearly a signalling colour. I guess that you are planning to publish another article on this subject and what to keep your powder dry, but it would would be good to have at least some comment on the generality of the phenomenon, what you see as its significance, and on the literature.

2. a) The literature in this area uses the terms: 'purity', 'saturation', and 'chroma' - and maybe even 'chromaticness', more or less vaguely and interchangeably. I think it would be useful to have an explicit definition of the metric used, **preferably supported by a diagram to help those readers not familiar with the details of pavo etc. **

b) For similar reasons it would be nice also to see how the could move in the lizard's receptor space.

c) No mention is made of 'luminance', yet this may well not be irrelevant to colour appearance, and should I think at least be documented -

d) Estimated receptor excitation for all colours studied should be documented and available to readers, if this is not already done.

3. Line 67. Structural colours are often not associated with a specialised 'iridophores' but the general chitin, keratin matrix of the epidmermis.

4. Line 98. The relevance/nature of 'allocated pigments' to structural coloration should be made clear, since the pigment if there is one is likely only to be melanin, which is not in short supply for these lizards
---

## [Author Response · Author response to Decision Letter 1]

30 Mar 2020

1. I am not aware of any other example of an animal showing such dramatic colour changes with temperature at least in a what is clearly a signalling colour. I guess that you are planning to publish another article on this subject and what to keep your powder dry, but it would would be good to have at least some comment on the generality of the phenomenon, what you see as its significance, and on the literature.

Response: We now acknowledge that dramatic cases of this may in fact be rare, when we discuss other, perhaps relatable, examples from the literature (line 457-467). We also address the potential significance of this trait as signal of thermal performance (lines 449-456).

2. a) The literature in this area uses the terms: 'purity', 'saturation', and 'chroma' - and maybe even 'chromaticness', more or less vaguely and interchangeably. I think it would be useful to have an explicit definition of the metric used, **preferably supported by a diagram to help those readers not familiar with the details of pavo etc. **

Response: Indeed, several interchangeable terms are found in the literature for the same property of color. To avoid confusion, we now refer to this metric exclusively as “saturation” throughout and elaborate on its meaning (lines 97-106).

b) For similar reasons it would be nice also to see how the could move in the lizard's receptor space.

Response: We considered including a figure illustrating the hue shifts in a color space. Unfortunately, the data was not as well represented as we hoped given their fine scale of our data in such a wide space. We have included the figure in a preliminary format here, and if you believe it would still be informative to the readers, we would be happy to include it in the manuscript.

c) No mention is made of 'luminance', yet this may well not be irrelevant to colour appearance, and should I think at least be documented -

Response: We have replicated our analyses of hue and saturation on mean brightness as well and provide a short description of its meaning (lines 340-345). With that, our manuscript now contains information on changes in hue, saturation, and brightness for this species.

d) Estimated receptor excitation for all colours studied should be documented and available to readers, if this is not already done.

Response: We now provide a table with mean ± SD excitation values for the four cone types, given for males and females and the two temperature treatments (lines 271-274).

3. Line 67. Structural colours are often not associated with a specialised 'iridophores' but the general chitin, keratin matrix of the epidmermis.

Response: We now address these other possibilities and relevant references (lines 68-69).

4. Line 98. The relevance/nature of 'allocated pigments' to structural coloration should be made clear, since the pigment if there is one is likely only to be melanin, which is not in short supply for these lizards.

Response: Since color is indeed such a diverse trait with a variety of underlying mechanisms across species, we opted to simply remove this statement altogether.

---

## [Editor Report · Decision Letter 2]

14 Apr 2020

PONE-D-19-26164R2

Plastic sexual ornaments: assessing temperature effects on color metrics in a color-changing reptile

PLOS ONE

Dear Mr Assis,

Thank you for submitting your manuscript to PLOS ONE. I am reluctantly asking for a minor revision, because I think that technically this ms does meet the PLoS ONE publication criteria,but there are couple of issues:

1. The cover letter states for comment 2b) b)'it would be nice also to see how the could move in the lizard's

receptor space.' that 'the data was not as well represented as we hoped given their fine scale

of our data in such a wide space. We have included the figure in a preliminary format

here....'  I do not see any such preliminary  figure, and I find the justification that the receptor space is too large odd, as it is easy to plot an enlarged fraction of the entire space. Although you argue for the use of  measure of colour (change) independent of the eye I think it is at least useful to know whether the colour changes that you document  a) their detectability; and b) their direction to a natural viewer.

2. Please comment that   in colour science saturation conventionally refer to an aspect of colour appearance (to human viewers) the usual definition of saturation,  i.e. difference from the most similar grey (Wyszecki & Stiles 1982; or similar elsewhere). The use of the term in the animal coloration literature to refer to some measure of variation in reflectance across the visible spectrum is now established, but somewhat confusing especially where colour appearance to the viewer is relevant.

3. It seems to me that your model 3, though somewhat arbitrary is at least quite general comapred to measures S1, and perhaps well matched to animal visual systems because rhodopsin spectral sensitivities have a half width of roughly 100nm. Some comment to this effect would be useful.

4. Are your spectra available as ESM or otherwise?  I think they should be.

As I say these are 'optional' comments and I will accept the ms with any reasonable final revisions.  This is a very interesting system and you have made a thoughtful and careful study.

We would appreciate receiving your revised manuscript by May 29 2020 11:59PM. To enhance the reproducibility of your results, we recommend that if applicable you deposit your laboratory protocols in protocols.io, where a protocol can be assigned its own identifier (DOI) such that it can be cited independently in the future. For instructions see: http://journals.plos.org/plosone/s/submission-guidelines#loc-laboratory-protocols

We look forward to receiving your revised manuscript.

Kind regards,

Daniel Osorio

Academic Editor

PLOS ONE

---

## [Author Response · Author response to Decision Letter 2]

27 Apr 2020

1. The cover letter states for comment 2b) b)'it would be nice also to see how the could move in the lizard's

receptor space.' that 'the data was not as well represented as we hoped given their fine scale

of our data in such a wide space. We have included the figure in a preliminary format

here....' I do not see any such preliminary figure, and I find the justification that the receptor space is too large odd, as it is easy to plot an enlarged fraction of the entire space. Although you argue for the use of measure of colour (change) independent of the eye I think it is at least useful to know whether the colour changes that you document a) their detectability; and b) their direction to a natural viewer.

Response: This time we submitted a figure that in our judgment was the most representative of hue and saturation changes in the lizard color space: a two-dimensional projection of the tetrahedral space from the perspective of the UV vertex (figure 3, lines 274-277)

2. Please comment that in colour science saturation conventionally refer to an aspect of colour appearance (to human viewers) the usual definition of saturation, i.e. difference from the most similar grey (Wyszecki & Stiles 1982; or similar elsewhere). The use of the term in the animal coloration literature to refer to some measure of variation in reflectance across the visible spectrum is now established, but somewhat confusing especially where colour appearance to the viewer is relevant.

Response: A statement referencing the definition of saturation in color science as well as our own in animal coloration was included along with the relevant references (Wyszecki & Stiles 1982; Wilkins & Osorio 2019) (lines 97-105; 440).

3. It seems to me that your model 3, though somewhat arbitrary is at least quite general comapred to measures S1, and perhaps well matched to animal visual systems because rhodopsin spectral sensitivities have a half width of roughly 100nm. Some comment to this effect would be useful.

Response: We mention the biological relevance of this approach due to the Gaussian distribution and sensitivity of cone sensory pigments (lines 410-412).

4. Are your spectra available as ESM or otherwise? I think they should be.

Response: We followed this suggestion and decided to upload all the raw spectra (.txt files) along with our dataset and R code in public repositories if this manuscript is accepted for publication.

As I say these are 'optional' comments and I will accept the ms with any reasonable final revisions. This is a very interesting system and you have made a thoughtful and careful study.

---

## [Editor Report · Decision Letter 3]

1 May 2020

Plastic sexual ornaments: assessing temperature effects on color metrics in a color-changing reptile

PONE-D-19-26164R3

Dear Dr. Assis,

We are pleased to inform you that your manuscript has been judged scientifically suitable for publication and will be formally accepted for publication once it complies with all outstanding technical requirements.

With kind regards,

Daniel Osorio

Academic Editor

PLOS ONE
---

## [Editor Report · Acceptance letter]

4 May 2020

PONE-D-19-26164R3 

Plastic sexual ornaments: assessing temperature effects on color metrics in a color-changing reptile 

Dear Dr. Assis:

I am pleased to inform you that your manuscript has been deemed suitable for publication in PLOS ONE. Congratulations! Your manuscript is now with our production department. 

With kind regards,

on behalf of

Dr. Daniel Osorio 

Academic Editor

PLOS ONE